# Bridging the Gap: Genetic Insights into Graft Compatibility for Enhanced Kiwifruit Production

**DOI:** 10.3390/ijms26072925

**Published:** 2025-03-24

**Authors:** Iqra Ashraf, Guido Cipriani, Gloria De Mori

**Affiliations:** Department of Agriculture, Food, Environmental and Animal Science, University of Udine, Via delle Scienze 206, 33100 Udine, Italy; ashraf.iqra@spes.uniud.it (I.A.); guido.cipriani@uniud.it (G.C.)

**Keywords:** *Actinidia*, graft union formation, primary and secondary metabolites, micrografting, rootstock/scion

## Abstract

Kiwifruit, with its unique flavor, nutritional value, and economic benefits, has gained significant attention in agriculture production. Kiwifruit plants have traditionally been propagated without grafting, but recently, grafting has become a more common practice. A new and complex disease called Kiwifruit Vine Decline Syndrome (KVDS) has emerged in different kiwifruit-growing areas. The syndrome was first recognized in Italy, although similar symptoms had been observed in New Zealand during the 1990s before subsequently spreading worldwide. While kiwifruit was not initially grafted in commercial orchards, the expansion of cultivation into regions with heavy soils or other challenging environmental conditions may make grafting selected kiwifruit cultivars onto KVDS-resistant or -tolerant rootstocks essential for the future of this crop. Grafting is a common horticultural practice, widely used to propagate several commercially important fruit crops, including kiwifruits, apples, grapes, citrus, peaches, apricots, and vegetables. Grafting methods and genetic compatibility have a crucial impact on fruit quality, yield, environmental adaptability, and disease resistance. Achieving successful compatibility involves a series of steps. During grafting, some scion/rootstock combinations exhibit poor graft compatibility, preventing the formation of a successful graft union. Identifying symptoms of graft incompatibility can be challenging, as they are not always evident in the first year after grafting. The causes of graft incompatibility are still largely unknown, especially in the case of kiwifruit. This review aims to examine the mechanisms of graft compatibility and incompatibility across different fruit crops. This review’s goal is to identify potential markers and techniques that could enhance grafting success and boost the commercial production of kiwifruit.

## 1. Introduction

The practice of grafting, in which the tissues of two plants are joined to allow them to continue growing together, has revolutionized fruit tree cultivation. This asexual propagation technique is considered the most efficient method for the propagation of fruit trees on a large scale, especially important for woody fruit trees like citrus, grapevines, apples, pears, peaches, and other *drupaceae*. It facilitates propagation, reduces fruiting time, and enhances resistance to biotic or abiotic stresses [1]. An important historical achievement occurred in the late 1800s when grapevine grafting was widely introduced [1]. Upon introducing the insect pest phylloxera to Europe from North America, European grape vines perished due to their lack of inherent resistance. An ingenious technique was employed in which grafting was used to substitute a susceptible European root with a disease-resistant North American one [1]. Hence, the technique of grafting American rootstocks onto European scions originated and is now employed in areas where phylloxera is prevalent, which constitutes almost all wine-producing regions worldwide.

Before the emergence of a complex syndrome that causes the collapse and death of most cultivars due to root system degeneration, the use of rootstock in kiwifruit cultivation was not widely used. Abiotic and biotic stresses determine the triggers of the plant disease known as KVDS. The future of kiwifruit cultivation in areas affected by KVDS relies on the availability of resistant or tolerant rootstocks. Research and experimentation by research institutes and private nurseries are moving towards using rootstocks obtained from wild species of the *Actinidia* genus. There is little information on this subject, but the demand from fruit growers is urgent. Similarly to many other fruit species, we must conduct a process to identify the factors that contribute to the success or failure of grafting methods.

Kiwifruit (*Actinidia* spp.) is a popular, nutritious fruit and an economically important crop grown in distinct parts of the world. It is a deciduous, dioecious vine belonging to the family *Actinidiaceae*. Kiwifruit has grown significantly as a horticultural crop during the last fifty years, first in New Zealand and later expanding to other countries like China and Italy [2]. Despite this worldwide expansion, two species, *A. chinensis* var. *deliciosa* and *A. chinensis* var. *chinensis*, dominate the kiwifruit market. Of these, the green-fleshed variety *A. chinensis* var. *deliciosa* ‘Hayward’ still accounts for 80% of the world’s total planted area. Following this, yellow-fleshed cultivars of *A. chinensis* var. *chinensis* were domesticated in China, significantly influencing the global kiwifruit industry. Today, the primary commercially available yellow-fleshed kiwifruit varieties outside China are ‘Gold 3-ZespriGold’ and ‘Jintao-Jingold’ [3]. Recently, red-fleshed *A. chinensis* var. *rufopulpa* cultivars have been introduced to the market [4,5]. New Zealand and Chile export more than 90% of their kiwifruit production, while Italy exports around 75% of its yield. In contrast, while Chile and New Zealand import minimal amounts, Italy imports around 5000 tons annually, especially during off-seasons [6]. Global demand for kiwifruit is increasing day by day. In recent years, climate change has introduced new challenges in managing kiwifruit orchards. Mismanagement of water resources has led to issues with waterlogging, which, combined with rising temperatures, has facilitated the emergence of a new and complex disease known as KVDS. This severe soil-borne disease affects the fine roots of *A. chinensis* var. *deliciosa* and *A. chinensis* var. *chinensis* and appears to result from interactions between various soil pathogens, primarily oomycetes, and waterlogging conditions. To tackle the problem of kiwifruit dieback, it will be crucial to revise agronomic practices for managing the entire orchard, including the adoption of resistant rootstocks for *Actinidia* to handle heavy soils and challenging conditions. The rootstocks from *A. chinensis* var. *chinensis* and *A. chinensis* var. *deliciosa* are commonly considered to be sensitive to waterlogging stress. Therefore, there is an urgent need to screen waterlogging-tolerant rootstocks and evaluate their effects on the waterlogging tolerance of grafted kiwifruit plants [7].

This review provides a comprehensive analysis of graft union formation, primarily in fruit trees, investigating the mechanisms that affect graft compatibility and incompatibility. It provides an in-depth insight into how genes, as well as primary and secondary metabolites, function in successful grafting across various fruit trees. Additionally, the review presents a new perspective on leveraging this knowledge to enhance grafting success in kiwifruits.

## 2. Grafting Techniques for Fruit Tree Propagation

Grafting techniques in fruit trees are essential for propagating desired cultivars. The most used grafting methods in fruit nurseries and orchards are bud grafting (which can be either chip-budding or T-budding), twin cleft whip grafting or tongue grafting, notch grafting, cleft grafting, ring and crown grafting (Figure 1). It is not known why a grafting technique works well in one species and poorly or not at all in others. Nurserymen apply different techniques based on empirical information, often jealously guarded. This knowledge is typically passed down through generations, creating a legacy of practices that evolve over time. As a result, successful grafting methods can vary significantly, even among closely related species, highlighting the intricate relationship between plant biology and horticultural techniques.

The grafting techniques used in kiwifruit cultivation include top grafting (tongue and cleft grafting), T-budding, and side grafting [8,9]. Tongue grafting has been used largely in the field (Figure 2). Celık and colleagues [10] conducted grafting experiments with the genotype “Hayward” cuttings on the 3-year-old seedling rootstock of the same variety in field conditions. The tested techniques include reverse T, T, chip bud, and machine and manual chip bud grafts. Their results demonstrated that manually applied chip bud grafts achieved the best outcomes with the highest graft retention rate (98.3%), graft application success (91.67%), graft shoot length (58.07 cm), and graft shoot diameter (6.84 mm). While the grafting machine used for chip bud grafting is effective, the process is time-consuming and has a lower graft success rate compared to the manual method. The cleft grafting method is considered the commercial standard for plum cultivation, with a high success rate of 9.67 out of 10 [11]. In walnuts, wedge grafting was found to be more successful than tongue grafting in terms of sprouting percentage, graft union success, and overall plant growth [12].

Another grafting technique, called “in vitro grafting or micro-grafting”, was introduced in the late 1990s, which is different from conventional grafting in several ways. The major difference is the smaller size of the scion and rootstock, which maintains aseptic conditions during the grafting procedure and graft union formation. Once in vitro grafting becomes successful, the grafted shoots can be cultured on the rooting medium for root development. This technique utilizes tissue culture-derived micro-scions and micro-rootstocks under sterile conditions [13]. In vitro grafting allows the researchers to control the grafting conditions more effectively and understand the scion-rootstock phenotypic changes in a precise manner. The micro-grafting technique has significant potential for plant improvement and large-scale propagation. One key advantage of this method is the production of virus-free plants, making it particularly useful in fruit crop propagation. Moreover, micro-grafting allows grafting operations to be performed at any time of the year. Due to its numerous benefits, this technology is practically applicable to researchers and nursery growers. Micro-grafting protocols have been developed for many fruit crops (Table 1), including grapes [14,15], kiwifruits [16,17,18], apples [19,20], and apricots [21].

## 3. Graft Union Formation in Different Fruit Trees: Compatibility and Incompatibility

Grafting involves connecting the shoot of one plant, termed the scion, onto the root of another plant, known as the rootstock. Successful grafting is based on establishing a vascular connection between the scion and rootstock [29]. The mechanism of successful grafting involves several processes, such as hormonal signaling, gene expression, plant metabolites, protein turnover, RNA silencing, and ion uptake and transport in grafted fruit trees [30,31]. Grafting success, termed “graft compatibility”, only occurs when both scion and rootstock belong to closely related taxonomic groups. During compatible graft union formation, the key steps are the adhesion of the rootstock and scion, the expansion of the callus or callus bridge, and the development of vascular connection across the graft interface [32,33]. However, in some cases, scion and rootstock combinations may not lead to successful graft union formation due to an incomplete genetic relationship between them, resulting in unsuccessful grafting termed “graft incompatibility” [34]. Graft incompatibility has been described in many fruit species and is generally referred to as the inability of the stock and scion to bind together to form a successful graft union [35]. Despite the increasing demand for quality planting materials for commercial production, farmers usually face the problem of poor graft success with huge losses. The main reason for graft incompatibility is the adverse physiological interactions between the scion and rootstocks, as well as anatomical irregularities at the graft junction, and this condition leads to the premature death of the grafted trees. The symptoms of graft incompatibility may not arrive immediately but can develop over time [36,37]. In the context of fruit tree cultivation, comprehending the genetic basis of graft compatibility and incompatibility is essential for enhancing grafting success rates and ensuring sustainable production.

Successful graft union formation is considered a strong determination of scion/rootstock compatibility, particularly in those species that are difficult to graft. The cellular processes involved in successful graft union formation are well understood and apply to both herbaceous and woody plants. However, the time required for successful graft union formation differs between most herbaceous and woody plants. For instance, fruit trees require several months to establish a successful graft interface [38,39].

The process of graft union formation in most fruit trees exhibits a high degree of similarities [29,35]. In spur-type apples, the histological progression of successful grafting is characterized by callus formation across all scion/rootstock combinations. At 90 DAG (ninety days after grafting), the establishment of the cambium and reconnection of vascular cells occurred. After the 120th DAG, the callus bridge fills the gap between the scion and rootstock and continues to develop for a few more days, leading to the establishment of the phloem and xylem [40]. Similarly, in cashews, graft union formation begins with the disappearance of the necrotic layer and the formation of callus at 30 DAG, followed by scion/rootstock adhesion at 60 DAG. By 98 DAG, vascular connection and healing processes were fully established [41]. In contrast, successful graft union formation in mango was observed earlier. According to histological analysis, callus formation began at 7 DAG, followed by continuous callus proliferation at 14 DAG, which facilitated the progression of the xylem and phloem. Additionally, vascular bundle differentiation was observed at 21 DAG [42].

Graft union formation in citrus has been investigated in field experiments; histological examination revealed that no anatomical differences were observed between compatible and incompatible interactions. Although compatible grafts exhibited a higher survival rate than incompatible ones, the healing process was similar in both cases [43].

In kiwifruit, graft union formation was observed during the micro-grafting process. Successful formation of vascular bundles was noted between 14 and 21 DAG. The scions used included three genotypes of *A. chinensis* var. *deliciosa* (‘Yuxiang’, ‘Jinmin’, and ‘Hayward’), and one genotype of *A. chinensis* var. *chinensis* (‘Qihong’), with *A. valvata* serving as the rootstock [17]. Micrografting in different combinations has been recently tested using *A. chinensis* var. *deliciosa* cv. ‘Hayward’, *A chinensis* var. *chinensis* cv. ‘KiKoKa’ grafted on themselves, as a control, and on genotypes of *A. valvata* and *A. macrosperma*. A solid connection between the tissues of the rootstocks and scions and the subsequent vigorous vegetative response took 10 days (Navacchi, personal communication) (Figure 3). Recent research has further elucidated the process of graft compatibility (Figure 4) and incompatibility in kiwifruit, demonstrating that compatible rootstock and scion combinations exhibit uniform cell structure and vascular connections at the graft junction. In contrast, incompatible pairings often display interruptions at the graft site, characterized by accumulations of necrotic cells and disorganized vascular structures [44]. Additionally, factors such as temperature, humidity, scion variety, rootstock, grafting time, and wrapping material influence successful graft union formation [45]. Understanding and optimizing these factors are crucial for improving graft success and enhancing the quality and productivity of kiwifruit cultivation.

## 4. Genetic Mechanism of Graft Compatibility/Incompatibility

Graft compatibility involves complex physiological and molecular interactions between the grafted partners. Recent studies have identified several genetic factors influencing these interactions.

### 4.1. Gene Expression Profile

Differential gene expression analysis between compatible and incompatible graft unions reveals critical regulatory pathways involved in vascular connection and wound healing processes. Only eight genes have been directly linked to graft formation, despite the widespread use of grafting to improve crops; most of these genes were found in *Arabidopsis thaliana* [33,46,47,48] (Table 2). Both temporal and rootstock/scion-specific molecular patterns linked to graft development have been clarified through transcriptomic characterization of junction formation [33,46,47,48,49]. Several plant species possess robust regeneration capabilities. Callus, a proliferative clump of cells, rapidly forms at cut locations and facilitates wound healing, even in cases of severe cuts in the stems. In a conceptual sense, a deep cut across a stem that severs the vascular tissue may be likened to a self-grafted plant, as the transcriptional responses to such wounds are comparable to those reported during graft formation [49,50]. Specifically, the activation of NAC DOMAIN-CONTAINING PROTEIN071 (*ANAC071*) and *ANAC096* expression was observed in both stem cutting and grafting. Mutations in these genes were found to impede both cutting and graft healing processes [50,51]. The regulation of wound healing, regeneration, and graft formation involves the participation of both WUSCHEL-LIKE HOMEOBOX13 (*WOX13*) and ETHYLENE RESPONSE FACTOR115 (*ERF115*) transcription factors [50,52,53]. Both *Physcomitrium* and *Marchantia* exhibit upregulation of *WOX13-like* and an *ERF115-like* homolog, *PpWOX13L* and *MpERF15*, in response to wounding. Disrupting these components hinders wound healing and response capabilities [54,55]. Furthermore, auxin is likely involved in the early recognition of wounds. Among the first genes activated during graft formation are several DNA binding with one finger (*DOF*) transcription factors. Zhang and colleagues [50] demonstrated that two *DOF* transcription factors, HIGH CAMBIAL ACTIVITY2 (*HCA2*) and TARGET OF MONOPTEROS6 (*TMO6*), were activated early in the grafting process, with *HCA2* playing a significant role in the healing process. In recent years, increasing attention has been given to studying gene expression changes during grafting to understand the mechanisms driving graft union formation in different fruit species (Table 2). Differentially expressed genes (DEGs) during graft union formation have been studied in *Carya* spp. These genes were involved in developmental processes such as auxin transport (*ARF*), signal transduction (*Type-B ARR*), cell cycle regulation (*Cyclin D-type CYCD*), ROS scavenging (*POD*, *CAT*, *APX*), metabolism (Cellulose synthase and LACCASE), programmed cell death (aspartic proteinase and ribonuclease), and potentially in cambium formation (*GA2ox*) and cellular growth (*NAC*) [56,57,58]. A study in grapevine showed that, in this case as well, many genes involved in cell wall synthesis, metabolism, cell organization, hormone signaling, and vascular development were upregulated at 3 and 28 DAG [59]. In pear, 14 known genes associated with kinase function, carbohydrate metabolism, protein metabolism, and cell activity and development were found to be involved in graft union formation [60]. Additionally, some studies have shown that grafting activates the plant’s antioxidant defense system; however, incompatible scion/rootstock combinations may lead to excessive production of ROS [61]. Specifically, at 10th DAG, compared to compatible grafts, the transcript level of six antioxidant genes (*SOD1*, *SOD3*, *APX3*, *APX6*, *CAT1*, and *CAT3*) was found to be suppressed in the incompatible heterografts [61].

In litchi, a study of compatible and incompatible graft interaction by RNA sequencing revealed 6060 and 5267 DEGs, respectively. Upregulated genes were involved in wound response and phenylpropanoid biosynthesis. The compatible graft interaction exhibited the upregulation of 9 DEGs related to the auxin pathway and 13 DEGs related to lignin biosynthesis, suggesting that these genes play a role in promoting graft compatibility and the healing process [62].

According to recent research on citrus graft combinations, DEGs between graft-compatible and incompatible interactions were linked to plant hormone synthesis and signal transduction (auxin (AX), abscisic acid (ABA), and ethylene-related genes). Their findings revealed that the balance between indole-3-acetic acid (IAA) and ABA biosynthesis and signaling plays a crucial role in graft compatibility [63]. Another investigation of graft incompatibility between lemons (*Citrus limon*) and sweet orange (*Citrus sinensis*) demonstrated DEGs associated with auxin biosynthesis [64].

Tomato (*Solanum lycopersicum*) and pepper (*Capsicum annuum*) grafted plants express signs of anatomical junction failure within the first week of grafting. By generating a detailed timeline for junction formation to pinpoint the cellular basis for this delayed incompatibility. Based on these important anatomical processes, which anticipate essential regulators for grafting, gene regulatory networks for compatible self-grafts and incompatible heterografts were deduced. Upon investigating the function of vascular development in graft formation, it was discovered that *SlWOX4* may regulate graft compatibility. It was discovered that *SlWOX4* homografts are unable to build xylem bridges across the junction after a functional investigation of this anticipated regulator. This indicates that *SlWOX4* is necessary for vascular reconnection during grafting and may serve as an early warning sign of graft failure [65].

**Table 2 ijms-26-02925-t002:** Summary of differentially expressed genes (DEGs) during grafting in different species.

Species	Functional Categories	Gene Classes Involved During Grafting	Reference
*Arabidopsis thaliana*	Wound-induced gene, Ethylene related genes, Transcriptional factors	*WOX13*, *ERF115*, *ANAC071*, *ANAC076*, *RAP2.6L*, *DOF*, *HCA2*, *TMO6*	[46,50,52,53]
Citrus (*Citrus maxima*, *Citrus limon*, *Citrus sinensis*)	Lignin biosynthesis, ROS scavenging, auxin-signal transduction, hormone signaling, ABA-related genes, ethylene-related genes, starch and sucrose metabolism, transcriptional factors	*CNGC1*, *PNP-A*, WR protein, *PEROX*, *EDS1*, *GA20OX2*, *LRR*, *WRKY51*, *SAUR*, *XCP1*, *HEXO2*, *NIMIN-2*, *TUB*, *NCEDs*, *PYL*, *PP2C*, *TIFY-10A*	[63,64]
Grapevines (*Vitis vinifera*)	Cell wall synthesis, secondary metabolism, vascular development hormone signaling	*F4P13*, *ENOD9*, *F6N15*, *CPs III*, *XCP1*, *Cuc*, *SAG101*, *APETALA2*, *P450*, *DEA1*, *PPFK*, *LRRIII*, *Myb-like 102*, *LEA14A*, *LBD4*, *LeDI-5c*, *AUX1-like*	[59]
Hickory (*Carya cathayensis*)	indole-3-acetic acid transport, signal transduction and mechanism, protein metabolism, water metabolism, and nuclear metabolism	*eIF-4A*, *CAT4*, *ARF*, K1/E32 K1 glycoprotein, *PIP1B*	[56]
Pecan (*Carya illinoinensis*)	hormone signaling, cell proliferation, xylem differentiation, cell elongation, cell wall deposition, programmed cell death, and ROS scavenging	*POD*, *CAT*, *APX*, *NAC*, Cellulose synthase, Laccase, Aspartic proteinase, Ribonuclease, *CYCD*, *ARF*, *Type-B ARR*, *GA2ox*	[57]
Litchi (*Litchi chinensis*)	metabolism, wound response, phenylpropanoid biosynthesis, auxin pathway, lignin biosynthesis	*TAA*, *YUC*, *Aux/IAA*, auxin-responsive protein IAA, auxin-induced protein, *PALs*, *C4Hs*, *C3H*, *4CLs*, *CCRs*, *COMTs*, *F5H*, *CADs*	[62]
Pear (*Pyrus communis* L.)/quince (*Cydonia oblonga*)	ROS scavenging	*APX1*, *APX2*, *APX3*, *APX6*, *CAT1*, *CAT3*, *Cu-ZnSOD1*, *Cu-ZnSOD2*, *Cu-ZnSOD3*	[61]
Pear (*Pyrus ussuriensis*)	kinase function, carbohydrate metabolism, protein metabolism, cell activity and development, nuclear metabolism, energy	*STY13-like*, *EIF-4A8*, *UBPS-like2B*, *FTSHI3*, *DEAD-box rde-12-like*, *MT-A70-like*, *pno1*, guanine nucleotide-binding protein 3, 7-deoxyloganetin glucosyltransferase-like, 2-hydrosylase FAH1-like, *At5g52850-like*, *CDC2*, *PHO1-like*, *ATPC*	[60]
Tomato (*Solanum lycopersicum*) and Pepper (*Capsicum annum*)	vascular development	*SlWOX4*	[65]

### 4.2. Hormonal Regulation

Phytohormones such as AX, cytokinin (CK), and gibberellin control the graft union processes and play a pivotal role in helping plants adapt to stress caused by grafting [33]. Some studies have shown that auxin produced by the vascular tissues of both the rootstock and scion plays a key role in inducing vascular tissue differentiation, acts as a plant growth regulator, and significantly contributes to graft union formation [66,67]. Moreover, lower levels of IAA promote the differentiation of phloem tissues, while higher concentrations stimulate the development of xylem tissues. In grafted plants, a significant shift occurs between the AX and CK [66]. This shift in balance contributes to the invigorating properties of the rootstock species, leading to enhanced graft growth rates. The underlying mechanism involves an increased supply of CK to shoot while auxin levels simultaneously decrease.

In contrast, ABA is a key factor in triggering dwarfing in higher plants. Indeed, dwarfing rootstocks of apple plants contain substantial amounts of ABA. A high concentration of ABA in the bark of dwarfing apple rootstock compared to vigorous apple rootstock is considered an efficient marker in rootstock selection [68,69]. The intriguing aim of grafting is to modify scion maturity through rootstock-induced changes. To revitalize growth, restore juvenile characteristics, and improve successful scion propagation, older scions from perennial plants with lengthy lifespans can be grafted onto younger rootstocks. Although the physiological and molecular mechanisms behind graft-induced rejuvenation are poorly understood, research on apples, *juglans*, and *pinus* suggests that increased auxin and, in certain situations, lower abscisic acid levels are linked to rejuvenation [70].

## 5. Genetic Distance

Greater genetic divergence between rootstock and scion can generally result in increased graft incompatibility. In horticulture, hetero grafting involves grafting two different genotypes to combine desirable traits from both the scion and the rootstock. In contrast, homo-grafting, where a plant is grafted with another plant of the same genotype, and auto-grafting, where a plant is grafted with itself, are primarily used in studies of fruit tree grafting. Among these, auto-grafting is expected to be consistently compatible due to their close taxonomic relationship [1].

Grafting within a species is generally successful; indeed, fruit trees that are closely related have a better chance of successful graft union formation compared to those that are more distantly related. For instance, a peach variety can typically be grafted onto any other peach variety as a rootstock. Moreover, grafting between species within a genus is usually successful; for example, peach and plum rootstock may be used for the commercial grafting of many varieties of almonds, apricots, and both European and Japanese plums [71,72,73]. Nicotiana can adhere to a wide variety of angiosperms by grafting. Cell wall reconstruction close to the graft interface is facilitated by a subclade of b-1,4-glucanases that are released into the extracellular space, according to comparative transcriptome investigations on graft combinations. The overexpression of b-1,4-glucanase encouraged grafting. Tomato fruits were produced on rootstocks from different plant families by using the Nicotiana stem as an inter-scion [48].

Additionally, recent studies in kiwifruit analyzed homo-grafts and hetero-grafts with various rootstock and scion combinations at 40, 80, and 120 DAG. Phenotypic and histological observations revealed that homo-graft combinations demonstrated high compatibility between scion and rootstock, whereas hetero-grafts showed increased localized graft incompatibility [44]. Hetero-grafting often results in increased expression of stress-responsive and secondary metabolism-related genes indicative of graft incompatibility, while homo-grafting and auto-grafting due to a closer genetic relationship shows higher compatibility.

A contrasting result is observed in citrus. Indeed, within the citrus family, Raiol-Junior and colleagues [74] assessed 86 rootstock/scion combinations from 18 species across 8 genera of the *Aurantioideae* subfamily. Their results revealed that 77% of these combinations were completely graft-incompatible [74]. Gene expression studies at the graft interface between homo-grafts and hetero-grafts were first conducted in grapevine by Cookson and co-workers [59]. They found that hetero-grafts exhibited higher expression of stress-responsive genes and had a better chance of successful graft union formation compared to homo-grafts [59].

## 6. Potential Markers Involved in Genetic Compatibility and Incompatibility in Fruit Trees

Identifying genetic markers associated with graft compatibility can facilitate the selection of compatible graft combinations.

### 6.1. Molecular Markers

In *Carya cathayensis*, a cDNA-AFLP analysis at 0, 3, 7, and 14 DAG identified several genes involved in IAA transport, cell cycle regulation, signal transduction, water metabolism, nuclear metabolism, amino acid metabolism, protein metabolism, carbon metabolism, and substance secretion. This detailed gene expression profiling highlights the dynamic and multifaceted nature of its physiological response to grafting [56]. Similarly, in *Pyrus ussuriensis* Maxim, cDNA-AFLP analysis identified genes involved in kinase activity, protein metabolism, carbohydrate metabolism, and other crucial cellular functions at different time points during the graft healing process [60].

Expanding on this line of research, QTL mapping in apricots has been used to identify genomic regions associated with graft incompatibility traits. Utilizing Next-Generation Sequencing (NGS) to generate SNP (single nucleotide polymorphism) markers, researchers have pinpointed specific loci that may influence the success of grafting in apricots [75]. This genomic approach provides a more targeted understanding of the genetic factors that govern graft compatibility, offering potential markers for breeding and selecting compatible graft combinations. The identification of allelic configurations and haplotypes of specific molecular markers that can be associated with a greater or lesser probability of success of a graft or of the appearance of symptoms determined by incompatibility phenomena is an objective that has not yet been fully achieved.

### 6.2. Transcriptomic and Proteomic Markers

As stated above, successful graft union formation involves multiple physiological, biochemical, and molecular changes. Transcriptomic and proteomic analyses have provided insights into the markers and mechanisms associated with successful grafting. Nocito et al., [76] observed that in the early stages of development, incompatible pear-quince hetero-grafts showed increased activity in five antioxidant enzymes: ascorbate peroxidase (APX), dehydroascorbate reductase (DHR), glutathione reductase (GR), superoxide dismutase (SOD) and catalase (CAT). Furthermore, specific isoforms of genes related to oxidative stress were identified in both pear-pear homo-grafts and pear-quince hetero-grafts [76].

Using gel electrophoresis-based proteomic analysis during callus grafting in *Prunus* spp., it was indicated that UDP-glucose pyrophosphorylase (UGPase) is a potential marker for graft compatibility. Lower UGPase expression was correlated with a decline in protein concentration and activity [77]. Moreover, proteomic analysis of graft incompatibility has revealed alterations in protein levels and gene expression related to enzymes involved in secondary metabolite production. These changes are often associated with the accumulation of secondary metabolites during the grafting process [78]. In *Prunus* spp., it has also been observed that at the graft interface, high expression levels of 4-coumarate ligase (4CL)—an enzyme involved in phenylpropanoid synthesis—are associated with graft incompatibility [79]. A different observation was made in grapevines. In this case, three 4CL genes showed higher expression at the graft interface of compatible hetero-grafts compared to the homo-grafted control from day 3 to day 28 after grafting [59].

## 7. Primary and Secondary Metabolites During Compatible and Incompatible Interaction

The compatibility and incompatibility of grafts are linked to various primary and secondary metabolites that are crucial for processes such as cell division, development, differentiation, and defense mechanisms [59,61]. For example, in the early stages of graft union formation in peach and plum, variations in sugar and starch levels can signal incompatibility issues [80]. After three months of grafting peach and plum, the levels of free amino acids and soluble proteins did not indicate a shortage of nitrogen or carbohydrates in the incompatible rootstock. Conversely, the scion showed patterns of nitrogen deficiency, as evidenced by the concentrations of asparagine, aspartate, and glutamate [81]. Moreover, although initial changes in primary metabolite profiles occur during graft union formation, carbon compound accumulation in the scion several months or years after grafting is often linked to graft incompatibility and disrupted phloem function [82,83].

Secondary metabolites, particularly phenolic compounds, play a significant role in determining graft compatibility and incompatibility in fruit trees. Research indicates that phenolic compounds accumulate at the graft interface, contributing to defense responses and various cellular processes, including cell division, development, and differentiation [84]. The synthesis of phenolic compounds begins with the conversion of the amino acid phenylalanine to ammonia and trans-cinnamic acid, catalyzed by the enzyme phenylalanine ammonia-lyase (PAL). Although *PAL* gene expression has been studied extensively during graft union formation and graft incompatibility, only one study conducted in grapevine has specifically examined PAL activity at the graft interface [82]. Moreover, pronounced variations in primary metabolites such as glucose, histidine, and lysine, as well as polyphenols, have been detected at the graft interface in grapevine compared to the surrounding woody tissues, with high concentrations of polyphenols but lower concentrations of certain flavanols, notably epicatechin [82,84]. Additionally, a higher concentration of gallic acid, catechin, and sinapic acid has been observed at the graft union in less compatible combinations, while lower levels of these compounds in more compatible combinations may be related to reduced oxidative stress, promoting better graft development [83].

In grafted *Uapaca Kirkiana* plants, an accumulation of p-coumaric acid was observed at the graft interface, and this accumulation may lead to non-differentiation of the tissue and degradation in the scion/rootstock interface, indicating graft incompatibility [85]. Further evidence suggests that grafting pear varieties onto quince rootstocks can result in the introduction of prunasin, a cyanogenic glycoside found naturally in quince but not in pears, into the pear’s phloem. prunasin, a well-known secondary metabolite linked to graft incompatibility in pear/quince combinations, is broken down in pear tissues by a glycosidase enzyme. This process releases toxic hydrocyanic acid at the graft junction, which can lead to cell death and damage to both xylem and phloem tissues, thereby causing graft incompatibility [86,87]. These studies provide valuable insight into the biochemical processes at the graft interface that influence graft compatibility and incompatibility in different fruit trees.

## 8. Trans-Grafting and siRNA Movement: Breakthrough or Barrier?

The exchange of signals, including hormones, DNA, RNA, microRNAs, and other molecules, between rootstocks and scions plays a crucial role in the growth and development of fruit plants [30,31]. Grafting onto rootstocks alters the levels of hormones such as cytokinins and auxins in both roots and shoots, promoting growth, postponing leaf aging, and enhancing tolerance to environmental stressors (Figure 5). This process has been extensively studied and has shown that hormonal changes play a significant role in the physiological responses of grafted plants, contributing to improved overall plant health and productivity.

Trans-grafting is a biotechnological technique where a genetically modified (GM) rootstock supports a non-GM (wild-type) scion, providing a protective advantage to woody plants. This approach has the potential to address consumer concerns regarding transgene flow. From a commercial perspective, using a single GM rootstock to propagate multiple cultivars could also reduce regulatory costs [88,89]. For instance, the transgenic expression of the mobile flower-promoting hormone blooming LOCUS-T (FT) is shown to be an effective method for inducing precocious blooming in blueberry and jatropha rootstock [90,91].

In plants, gene silencing signals carried by small interfering RNAs (siRNAs) can propagate locally between cells and travel long distances through the phloem. siRNAs are linked to the transmission of these silencing signals via plasmodesmata, the cytoplasmic channels connecting plant cells [92]. In this context, RNAi-based rootstocks are expected to efficiently transfer silencing molecules to non-transformed scions and can be used to develop virus-resistant transgenic plants [93]. Compatibility is crucial for the successful transmission of RNAi-silencing signals into the scion and the initiation of systemic gene silencing [66]. In tobacco plants, it has been observed that mobile siRNA from the rootstock triggered transcriptional gene silencing (TGS) in scion. Conversely, when the scion served as the siRNA donor, strong gene silencing was observed in the roots [94]. In sweet cherry, it has been reported that siRNA molecules derived from a hairpin gene construct can spread between cells and travel systemically over long distances. This ability to reach up to 1.2 m from the graft union is crucial for understanding the extent of systemic transmission [95].

Further research conducted on Arabidopsis has shown that the deliberate upregulation of microRNA156, a microRNA that promotes the youthful growth of plants, can effectively restore certain characteristics of juvenile plants in mature plants. However, the findings of a recent study on avocado indicate that while rootstocks can influence scion precocity, the expression of age-related molecular markers, such as microRNA156, is primarily regulated by the scion rather than the rootstock [35].

Studies on grapevine have demonstrated the production of virus-resistant GM rootstock through the expression of virus-derived proteins, such as the protein derived from *Plum Pox Virus* (PPV). However, these studies did not explain whether the resistance could be systemically transmitted from the GM rootstock to a wild-type scion [96]. Indeed, recent studies in plums have shown that RNAi signals, which are responsible for gene silencing, were not successfully transmitted across the graft junction, hindering the transfer of genetic information. The failure to achieve PPV resistance was due to the insufficient transfer of transgene-derived silencing signals from GM rootstock to scion. Although minor translocation of siRNA was detected in some samples, it was inadequate to induce resistance against viral attack [97]. Likewise, the transfer of transgenic siRNA from the graft to non-transformed leaves in walnuts was also not observed [98].

Another study in apple demonstrated that RNAi silencing signals, crucial for gene regulation, were not successfully transmitted across grafts in greenhouse experiments, highlighting challenges in genetic communication.

The authors hypothesize that lignification of both the scion and rootstock during the transfer from the in vitro to the greenhouse may have impeded cell-to-cell transport of siRNAs. These observations suggest that the success of RNAi-based technology depends on several factors, such as the level of siRNA expression, its mobility, and its interaction with the target gene. The observed variability in results is likely due to the differences in plant species, transgenic constructs, and targeted sequences [99]. However, trans-grafting and siRNA movement remain theoretical without extensive validation in practical horticultural contexts.

## 9. Conclusions and Future Perspectives

Grafting is a long-standing method that has existed for more than a thousand years. Its effectiveness in contemporary agriculture can be attributed to both living and non-living pressures, enhancing crop productivity and ensuring optimal results for farmers from their planted crops. This review provides a comprehensive analysis of the grafting mechanism and highlights the critical role of genetic compatibility and incompatibility in fruit trees, including kiwifruit. It explored the essential cellular and molecular processes involved in successful graft union formation. Identifying potential markers and developing effective grafting techniques are crucial for advancing genetic research and improving kiwifruit breeding programs. To deepen the understanding and application of grafting in kiwifruit cultivation, future research should focus on comprehensive genomic and proteomic analysis to identify the additional genes involved in genetic compatibility and incompatibility. This will enhance our understanding of the mechanisms that promote successful grafting and help identify potential markers for selecting compatible rootstock combinations. Establishing reliable markers for early detection of graft compatibility. Advanced grafting techniques such as micrografting will facilitate the cultivation of kiwifruit by investigating the use of controlled environmental conditions and protocols. By addressing these areas, future research can significantly improve the efficiency and success of kiwifruit grafting, contributing to the sustainability and profitability of kiwifruit cultivation and global kiwifruit production.

## Figures and Tables

**Figure 1 ijms-26-02925-f001:**
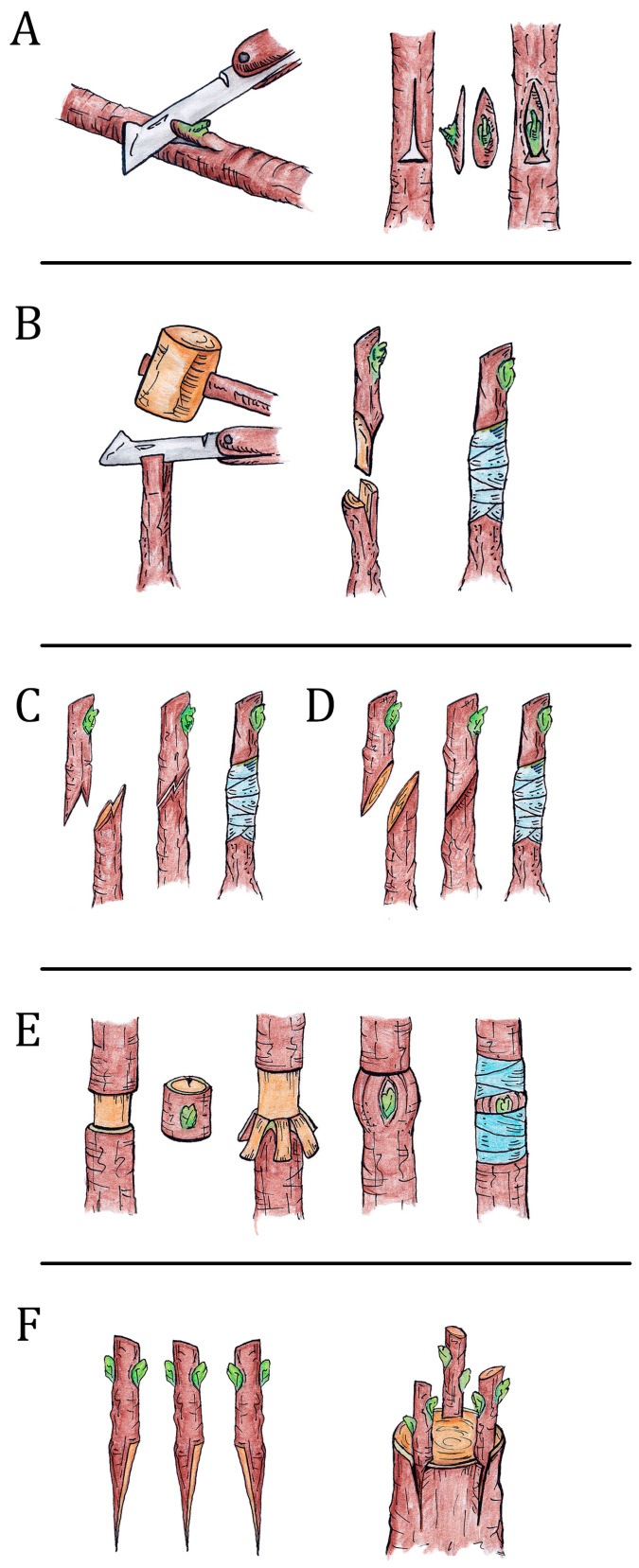
Graphical representation of different types of grafting (G. De Mori drawing). (**A**). Bud-grafting; (**B**). cleft grafting; (**C**). twin cleft whip grafting; (**D**). splice grafting; (**E**). ring grafting; (**F**). crown grafting.

**Figure 2 ijms-26-02925-f002:**
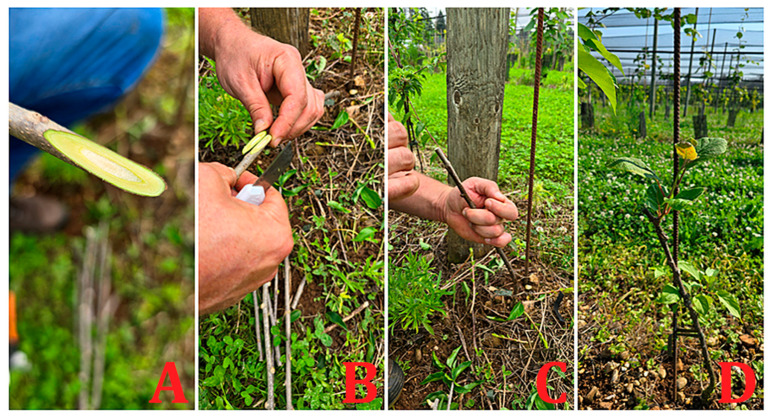
Field grafting in kiwifruit: (**A**) whip-cutting preparation; (**B**) approximation of rootstock and scion; (**C**) connection and binding; (**D**) vegetative development of the variety’s bud (scion).

**Figure 3 ijms-26-02925-f003:**
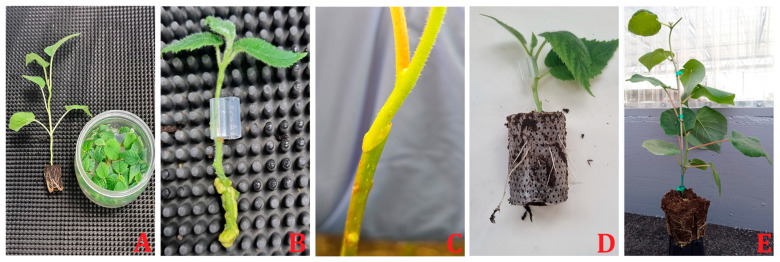
Micrografting steps in kiwifruit. **(A**) In vitro micropropagation of rootstock and scion plantlets; (**B**) micrografting; (**C**) micrografting results in a solid connection between the tissues of the rootstocks and scions after 10 days; (**D**) rooted micrografted plantlet; (**E**) First-year development of micrografted plant.

**Figure 4 ijms-26-02925-f004:**
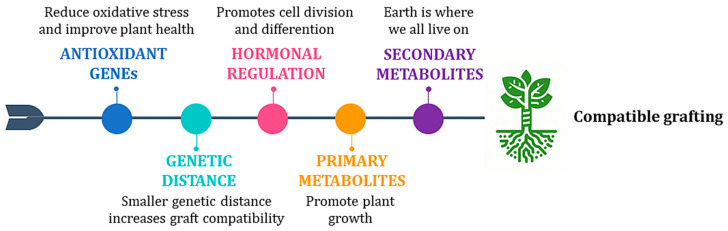
Illustration of a compatible graft in kiwifruit. Graft compatibility results from the synergy of multiple factors: activation of genes involved in oxidative processes, genetic distance between scion and rootstock, hormonal regulation, and the involvement of primary and secondary metabolites.

**Figure 5 ijms-26-02925-f005:**
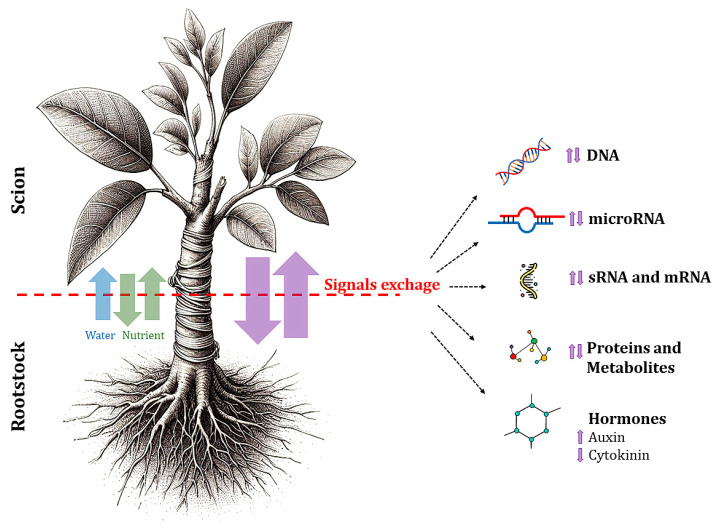
Graphical representation of the signal exchanges occurring between rootstock and scion.

**Table 1 ijms-26-02925-t001:** Success rate and applications of in vitro grafting techniques for various plant species.

Species	In Vitro Grafting Technique	Success Rate	Purpose of Study	Reference
Almond	Slit and wedge grafting	100%	Develop a micrografting system for mass commercial production of selected cultivars	[22]
Apple	Vertical slit wedge and horizontal	35.76%	Access the potential use and applicability of micrografting to develop invitro grafted plantlets	[19]
	Wedge (V-shape)	95%	Develop an efficient grafting method for the commercial production of apple	[20]
Apricot	Callus grafting	60–70%	To identify the graft incompatibility at an early stage	[21]
Citrus	Invert T-grafting	75%	Validate multiple parameters in shoot tip grafting	[23]
	Tip grafting	30–50%	Production of Indian ringspot virus-free plants	[24]
Grapevine	Wedge (V-shape)	70–90%	Develop an in vitro micrografting procedure for grapevines to facilitate rapid diagnosis of grapevine corky bark	[15]
		75–85%	Screen for virus-tolerant graft combinations	[14]
Kiwifruit	Cleft grafting	50–73%	Develop an efficient grafting method to produce virus-free seedlings	[18]
	V-shape grafting	100%	Evaluation of graft-compatibility at early stage	[17]
		100%	Evaluation of drought tolerance in micrografts	[16]
Mandarin Orange	Cleft and invert T graft	57%	Production of Citrus Tristeza virus-free plants	[25]
Pistachio	Wedge	95%	Develop micrografting protocol	[26]
Sour Cherry and Plum	Wedge (V-shape)	88–96%	Study virus–host adaptation for plum pox virus	[27]
Walnut	Vertical slit	90%	Test multiple parameters of micrografting	[28]

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
