# Peer review of "Bridging the Gap: Genetic Insights into Graft Compatibility for Enhanced Kiwifruit Production"

_ijms, 2025, doi:10.3390/ijms26072925_

Round 1
Reviewer 1 Report
Comments and Suggestions for Authors
The review article titled "Bridging the Gap: Genetic Insights into Graft Compatibility for Enhanced Kiwifruit Production" presents a comprehensive overview of the mechanisms underlying graft compatibility and incompatibility in fruit trees, specifically focusing on kiwifruit (Actinidia spp.). The integration of genetic understanding, metabolomic profiling, and refined grafting techniques is crucial in addressing the needs of modern agriculture and ensuring sustainability in kiwifruit cultivation.
In my opinion, this is:
- a thorough review of the literature on grafting in fruit trees, including the historical background, grafting techniques, and recent advancements in micrografting;
- a complex concept of graft incompatibility and graft union formation explained clearly, making it accessible to readers from various backgrounds;
- a strong effort to emphasize the importance of grafting in kiwifruit production, highlighting the need for resistant rootstocks to tackle the Kiwifruit Vine Decline Syndrome (KVDS) challenge.
However, in my opinion, the article has several weaknesses:
- the article is divided into several sections, which, while informative, can make it challenging to follow the author's arguments and main points;
- lacking original research; the review article relies heavily on existing literature, with a very few new findings or original research contributions;
- nearly half of the overall references are from outdated literature (published at least 10 years ago). This is particularly concerning in the context of Sections 6 to 11, where recent developments are being discussed.
Minor issues and some minor polishing of the English is needed.
The Abstract
The sentence "the syndrome was first recognized in Italy, although similar symptoms were observed in the 1990s in New Zealand, and then spreading worldwide" could be made clearer. The authors could consider rephrasing for clarity: "The syndrome was first recognized in Italy, although similar symptoms had been observed in New Zealand during the 1990s before subsequently spreading worldwide."
The use of "its goal" in the last sentence could be revised to clarify that "this review's goal" refers specifically to the review being discussed, improving the coherence of the text.
The authors stated that the "Grafting methods and genetic compatibility are crucial factors that affect fruit quality, yield, environmental adaptability, and disease resistance". Instead, for improved sentence structure, I suggest more concise option: "Grafting methods and genetic compatibility crucially affect fruit quality, yield, environmental adaptability, and disease resistance."
1. The Introduction
In the sentence "An important historical achievement occurred in the late 1800s when grapevine grafting was widely first introduced [1]", the phrase 'widely first introduced' can be rewritten for better clarity and grammar as 'widely introduced for the first time'.
The writing style is generally clear and concise, but some sentences could be rephrased for better clarity. For example, the sentence "The survival of kiwifruit cultivation in areas affected by KVDS will depend on the availability of resistant or tolerant rootstocks" could be rephrased as "The future of kiwifruit cultivation in areas affected by KVDS relies on the availability of resistant or tolerant rootstocks".
3. Graft Compatibility and Incompatibility
The text the authors provided is mostly well-structured and scientifically sound; however, it could benefit from minor adjustments to enhance clarity and precision.
I suggest the rephrase of the last sentence “In fruit tree cultivation understanding the genetic basis of graft compatibility and incompatibility is essential for improving grafting success rates”, into “In the context of fruit tree cultivation, comprehending the genetic basis of graft compatibility and incompatibility is essential for enhancing grafting success rates and ensuring sustainable production.
4. Grafting Techniques in Fruit Trees
The Section title "Grafting Techniques in Fruit Trees" could be rephrased into the "Grafting Techniques for Fruit Tree Propagation".
5. Graft Union formation in different fruit trees
As a suggestion, in the sentence "Ninety days after grafting (DAG), the establishment of the cambium and reconnection of vascular cells occurred," it would be better to use "at 90 DAG" instead of "Ninety days after grafting (DAG)" for consistency.
7. Hormonal Regulation
Minor Errors (line 290): "invigorating properties of rootstock species" should be "invigorating properties of THE rootstock species".
Instead of "ABA is one of the main factors responsible for triggering the process of dwarfing in higher plants", the sentence could be rephrased for clarity, e.g., "ABA is a key factor in triggering dwarfing in higher plants."
9. Potential markers involved in genetic compatibility and incompatibility in fruit trees
9.1. Molecular markers
9.2. Transcriptomic and proteomic markers
Please correct the term “peer quince hetero-grafts” (line 364) into “pear quince hetero-grafts”.
12. Conclusion and Future Perspectives
The text is generally free of grammatical errors, but I suggest some minor adjustments which could improve clarity:
Line 486: "Its presence in contemporary agriculture is attributed to its effectiveness" could be rephrased as "Its effectiveness in contemporary agriculture can be attributed"
Line 488: "guaranteeing that farmers achieve optimal results" could be rephrased as "ensuring optimal results for farmers".
Overall, this is a comprehensive study, and the authors did a great job. The article contribution reflects in highlighting the urgent need for research on grafting as a solution to combat Kiwifruit Vine Decline Syndrome (KVDS), which poses a significant threat to kiwifruit production. The article synthesizes genetic, hormonal, and molecular interactions that govern graft compatibility, offering valuable insights for both theoretical and practical applications in horticulture. Additionally, it identifies potential genetic markers that could improve graft selection and success rates, paving the way for enhanced kiwifruit productivity.
However, one potential weakness is that, while the discourse on genetic markers is promising, the article may lack specific actionable strategies or methodologies for implementing these findings in practical settings. Furthermore, the discussion on advanced techniques such as trans-grafting and siRNA movement remains theoretical without extensive validation in practical horticultural contexts.
Accordingly, I recommend “major revision” for its publication in the “International Journal of Molecular Sciences”.
Comments on the Quality of English Language
Some minor polishing of the English is needed.
Author Response
The authors thank the reviewers for their suggestions to improve the manuscript.
We respond to each single point below. All the changes made in the text have been highlighted in yellow. The bibliography has been renumbered in accordance with the text adjustments and the merging of some sections, as suggested by Reviewer 1.
Reviewer 1
Comments and Suggestions for Authors
The review article titled "Bridging the Gap: Genetic Insights into Graft Compatibility for Enhanced Kiwifruit Production" presents a comprehensive overview of the mechanisms underlying graft compatibility and incompatibility in fruit trees, specifically focusing on kiwifruit (Actinidia spp.). The integration of genetic understanding, metabolomic profiling, and refined grafting techniques is crucial in addressing the needs of modern agriculture and ensuring sustainability in kiwifruit cultivation.
In my opinion, this is:
- a thorough review of the literature on grafting in fruit trees, including the historical background, grafting techniques, and recent advancements in micrografting;
- a complex concept of graft incompatibility and graft union formation explained clearly, making it accessible to readers from various backgrounds;
- a strong effort to emphasize the importance of grafting in kiwifruit production, highlighting the need for resistant rootstocks to tackle the Kiwifruit Vine Decline Syndrome (KVDS) challenge.
However, in my opinion, the article has several weaknesses:
- the article is divided into several sections, which, while informative, can make it challenging to follow the author's arguments and main points;
We acknowledge the reviewer’s feedback and we have merged some sections of the manuscript to simplify its structure, as suggested.
- lacking original research; the review article relies heavily on existing literature, with a very few new findings or original research contributions;
We have conducted a thorough bibliographical research on the topic of our review and, as the reviewer correctly points out, we have not identified ground-breaking discoveries. This aligns with the current state of knowledge, particularly regarding kiwifruit, where the use of rootstocks has so far been episodic and quite limited. We believe we have provided a comprehensive overview without overlooking key information in the topics we have considered. Additionally, our group is actively engaged in research and testing of kiwifruit rootstocks, recognizing their increasing relevance in response to recent emerging challenges.
- nearly half of the overall references are from outdated literature (published at least 10 years ago). This is particularly concerning in the context of Sections 6 to 11, where recent developments are being discussed.
As mentioned in our response to the previous reviewer's comment, we have made every effort to ensure that no important information has been overlooked in the sections of our manuscript. While we acknowledge that some references may appear dated, we deliberately included them to provide appropriate citations for foundational knowledge and key background information relevant to our review. Our aim was to present a comprehensive overview that reflects both historical insights and current advancements in the field.
Minor issues and some minor polishing of the English is needed.
We thank the reviewer for carefully reading the manuscript and suggestions to improve the understanding of English. We have accepted all the suggestions that were made to us.
The Abstract
The sentence "the syndrome was first recognized in Italy, although similar symptoms were observed in the 1990s in New Zealand, and then spreading worldwide" could be made clearer. The authors could consider rephrasing for clarity: "The syndrome was first recognized in Italy, although similar symptoms had been observed in New Zealand during the 1990s before subsequently spreading worldwide."
The use of "its goal" in the last sentence could be revised to clarify that "this review's goal" refers specifically to the review being discussed, improving the coherence of the text.
The authors stated that the "Grafting methods and genetic compatibility are crucial factors that affect fruit quality, yield, environmental adaptability, and disease resistance". Instead, for improved sentence structure, I suggest more concise option: "Grafting methods and genetic compatibility crucially affect fruit quality, yield, environmental adaptability, and disease resistance."
- The Introduction
In the sentence "An important historical achievement occurred in the late 1800s when grapevine grafting was widely first introduced [1]", the phrase 'widely first introduced' can be rewritten for better clarity and grammar as 'widely introduced for the first time'.
The writing style is generally clear and concise, but some sentences could be rephrased for better clarity. For example, the sentence "The survival of kiwifruit cultivation in areas affected by KVDS will depend on the availability of resistant or tolerant rootstocks" could be rephrased as "The future of kiwifruit cultivation in areas affected by KVDS relies on the availability of resistant or tolerant rootstocks".
- Graft Compatibility and Incompatibility
The text the authors provided is mostly well-structured and scientifically sound; however, it could benefit from minor adjustments to enhance clarity and precision.
I suggest the rephrase of the last sentence “In fruit tree cultivation understanding the genetic basis of graft compatibility and incompatibility is essential for improving grafting success rates”, into “In the context of fruit tree cultivation, comprehending the genetic basis of graft compatibility and incompatibility is essential for enhancing grafting success rates and ensuring sustainable production.
- Grafting Techniques in Fruit Trees
The Section title "Grafting Techniques in Fruit Trees" could be rephrased into the "Grafting Techniques for Fruit Tree Propagation".
- Graft Union formation in different fruit trees
As a suggestion, in the sentence "Ninety days after grafting (DAG), the establishment of the cambium and reconnection of vascular cells occurred," it would be better to use "at 90 DAG" instead of "Ninety days after grafting (DAG)" for consistency.
- Hormonal Regulation
Minor Errors (line 290): "invigorating properties of rootstock species" should be "invigorating properties of THE rootstock species".
Instead of "ABA is one of the main factors responsible for triggering the process of dwarfing in higher plants", the sentence could be rephrased for clarity, e.g., "ABA is a key factor in triggering dwarfing in higher plants."
- Potential markers involved in genetic compatibility and incompatibility in fruit trees
9.1. Molecular markers
9.2. Transcriptomic and proteomic markers
Please correct the term “peer quince hetero-grafts” (line 364) into “pear quince hetero-grafts”.
- Conclusion and Future Perspectives
The text is generally free of grammatical errors, but I suggest some minor adjustments which could improve clarity:
Line 486: "Its presence in contemporary agriculture is attributed to its effectiveness" could be rephrased as "Its effectiveness in contemporary agriculture can be attributed"
Line 488: "guaranteeing that farmers achieve optimal results" could be rephrased as "ensuring optimal results for farmers".
Overall, this is a comprehensive study, and the authors did a great job. The article contribution reflects in highlighting the urgent need for research on grafting as a solution to combat Kiwifruit Vine Decline Syndrome (KVDS), which poses a significant threat to kiwifruit production. The article synthesizes genetic, hormonal, and molecular interactions that govern graft compatibility, offering valuable insights for both theoretical and practical applications in horticulture. Additionally, it identifies potential genetic markers that could improve graft selection and success rates, paving the way for enhanced kiwifruit productivity.However, one potential weakness is that, while the discourse on genetic markers is promising, the article may lack specific actionable strategies or methodologies for implementing these findings in practical settings. Furthermore, the discussion on advanced techniques such as trans-grafting and siRNA movement remains theoretical without extensive validation in practical horticultural contexts.
We appreciate the reviewer’s comments. In response, we have introduced a short paragraph to the molecular markers section to address the reviewer's valid observation regarding the need for practical implementation strategies. Additionally, we also included a sentence in the siRNA movement section acknowledging that, as the reviewer correctly points out, the practical application of this technique remains limited in most fruit species. We believe these additions provide better context and clarity regarding the current state of these approaches in horticultural practice.
Accordingly, I recommend “major revision” for its publication in the “International Journal of Molecular Sciences”.
We have done our best to improve our manuscript and we greatly appreciate the work that has been done by the reviewer to point out how to improve the text.
Comments on the Quality of English Language
Some minor polishing of the English is needed.
Reviewer 2
Comments and Suggestions for Authors
The review aims to describe update on genetic, physiological, hormonal processes during graft propagation of woody plants with purpose to propose future opportunities and necessary research of kiwifruit propagation. The manuscript is written in good language and easy to understand.
My suggestion is to improve images in Figure 3. They seem two different images illustrating different ideas. I suggest to separate them. The design of the first drawing leads to assumption that this is timeline, like, the process starts with antioxidant gene (activation?), then goes forward up to secondary metabolites (synthesis?). However, that’s not true – there are simultaneous processes ensuring graft compatibility. I suggest to redesign it.
We have taken up the reviewer’s suggestion to improve the understanding of what we want to represent in figure 3 and have produced two independent figures (2 and4).
Further - the photos of the graft process - should be referred to text more directly. And yes - why not show in vitro grafting then, that is quite new method and not everyone might be familiar with it? The showed method is rather old and well known, and widely used elsewhere. If You decide to leave these images, then describe also the traditional grafting methods, used for kiwifruit.
We have included additional photographs to illustrate the micrografting technique generating Figure 3, as suggested by the reviewer.

Reviewer 2 Report
Comments and Suggestions for Authors
The review aims to describe update on genetic, physiological, hormonal processes during graft propagation of woody plants with purpose to propose future opportunities and necessary research of kiwifruit propagation. The manuscript is written in good language and easy to understand.
My suggestion is to improve images in Figure 3. They seem two different images illustrating different ideas. I suggest to separate them. The design of the first drawing leads to assumption that this is timeline, like, the process starts with antioxidant gene (activation?), then goes forward up to secondary metabolites (synthesis?). However, that's not true - there are simultaneous processes ensuring graft compatibility. I suggest to redesign it.
Further - the photos of the graft process - should be referred to text more directly. And yes - why not show in vitro grafting then, that is quite new method and not everyone might be familiar with it? The showed method is rather old and well known, and widely used elsewhere. If You decide to leave these images, then describe also the traditional grafting methods, used for kiwifruit.
Author Response

(The authors gave the same response as above.)

Round 2
Reviewer 1 Report
Comments and Suggestions for Authors
The review article titled "Bridging the Gap: Genetic Insights into Graft Compatibility for Enhanced Kiwifruit Production" provides a comprehensive overview of the mechanisms that govern graft compatibility and incompatibility in fruit trees, focusing specifically on kiwifruit (Actinidia spp.). The authors have significantly improved the manuscript's readability, polished the language, and provided original research contributions, although the cited references remain unchanged.
Overall, this study is thorough, and the authors have done an excellent job. In my opinion, the manuscript merits publication in its current form in the “International Journal of Molecular Sciences.”